# Impact of Menopausal Symptoms on Work: Findings from Women in the Health and Employment after Fifty (HEAF) Study

**DOI:** 10.3390/ijerph20010295

**Published:** 2022-12-24

**Authors:** Stefania D’Angelo, Gregorio Bevilacqua, Julia Hammond, Elena Zaballa, Elaine M. Dennison, Karen Walker-Bone

**Affiliations:** 1MRC Lifecourse Epidemiology Centre, University of Southampton, Southampton SO15 3BX, UK; 2MRC Versus Arthritis Centre for Musculoskeletal Health and Work, University of Southampton, Southampton SO15 3BX, UK; 3Monash Centre for Occupational and Environmental Health, Monash University, Melbourne, VIC 3004, Australia

**Keywords:** menopause, working women, psychosocial work environment, physically demanding work

## Abstract

Women make up a growing proportion of the workforce and therefore many women experience menopause while in paid employment. We explored the prevalence of menopausal symptoms, the relationship between symptoms and coping with work and the risk factors associated with struggling at work during the menopause. The Health and Employment After Fifty (HEAF) community-based cohort of people aged 50–64 years was incepted 2013–2014 to study health and work. In 2019, female participants were asked to complete a questionnaire about their menopausal symptoms, and effect of those symptoms on their ability to cope at work. 409 women were eligible for inclusion. The commonest symptoms were vasomotor (91.7%); trouble sleeping (68.2%); psychological (63.6%) and urinary (49.1%). The prevalence of reporting symptoms was similar no matter which type of occupation women were performing at the time. Around one-third of women reported moderate/severe difficulties coping at work because of menopausal symptoms. Risk factors for difficulties coping at work included: financial deprivation, poorer self-rated health, depression, and adverse psychosocial occupational factors but not physical demands. More awareness is needed amongst employers in all sectors but women with financial difficulties and those with jobs in which they feel insecure, unappreciated, or dissatisfied are at greatest risk.

## 1. Introduction

In many countries, the participation of women in the work force has increased rapidly in one or two generations [1]. In the Organization for Economic and Commercial Development (OECD) countries, 60% of women are in paid work [2]. Some women may return to the workforce in the fifth, sixth or seventh decade of life as family commitments allow or for financial reasons [3], and many women now remain in work (and increasingly full-time work) throughout their working lives. For example, data from the UK show that women now make up 45% of the workforce aged >50 years [1]. Contemporaneously, many countries are applying policies which require people to work to older ages [4]. Consequently, an increasing number of women will be in paid work when they experience the menopause, typically aged between 45–55 years (average age 51 years). Not all women experience a symptomatic menopause, but some women report severe, debilitating symptoms, with the possibility of causing marked impact on their ability to function productively in the workplace, or even remain in paid work [5,6].

Despite the potential impact of menopause on women’s working lives, this research area had received scant attention until the last decade. Since then, studies have shown that the self-reported work ability of women, as assessed using the Work Productivity Activity Impairment scale [7], is affected to the greatest extent amongst those women who report more severe menopausal symptoms, as compared with women without symptoms [8]. Higher rates of sickness absence and productivity loss resulting in an increased indirect cost for employers have also been demonstrated amongst women with severe menopausal symptoms [2,8,9,10,11]. It has also been reported that women with severe menopausal symptoms felt less motivated at work and that this provokes some women to consider changing jobs, reducing their working hours or leaving the workforce altogether [12,13].

Some studies have explored the effects of different types of menopausal symptoms on work. For example, irritability and mood changes were found to have the worst impacts on job performance in one study of older female employees, although less so among women who were managers [14]. Another study found that vasomotor symptoms were associated with impaired work ability, after adjustment for demographic and lifestyle factors [15], but a larger study including women from Europe, Japan and the USA found that vasomotor symptoms appeared to cause greater impact on daily activities than working activities [16].

Whilst these studies contribute useful information, they shed no light on whether menopausal symptoms affect women doing different types of work in the same or different ways. The few studies that have considered work characteristics have tended to focus only on one type of worker, namely teachers [17], nurses [18,19], women employed by a university [20] and women working in professional, managerial and administrative (non-manual) occupations [21]. It is plausible that the burdensomeness of some symptoms might differ depending on the nature of work demands for example urinary symptoms with physically demanding work. To investigate this hypothesis therefore, we used data from a contemporary cohort of a population sample of British women who were working in a range of different occupations when they experienced the menopause to investigate: the prevalence of menopausal symptoms; the proportion who experienced difficulties at work relating to those symptoms; and which factors (socio-economic, lifestyle, occupational and nature and number of menopause symptoms) were associated with reporting problems coping with menopausal symptoms while at work.

## 2. Methods

This was a cross-sectional analysis of data collected as part of the Health and Employment After Fifty (HEAF) study, the detailed methodology of which has been previously reported [22]. Briefly, the sampling frame for this cohort study were 24 English General Practices geographically spread across England and representing every decile of social deprivation according to the 2010 English Index of Multiple Deprivation [22,23]. Participating practices mailed an information leaflet, consent form and baseline questionnaire to adults aged 50–64 years on their register (after excluding any whom it was deemed insensitive to contact). Willing participants returned their questionnaire and gave consent for annual follow-up thereafter. Each questionnaire collected information about demographics, employment status, psychosocial and physical job characteristics, lifestyle factors, and finances. In the fifth follow-up questionnaire (2019), a section focusing on menopause and reproductive health was added for female participants: women were asked about their reproductive history, contraception, menopausal status, age at starting menopause. Women were also asked to report whether they experienced or not any of the following menopausal symptoms: trouble sleeping, palpitations and panic; psychological (anxiety, depression, irritability, tearfulness); forgetfulness; vasomotor (hot flushes, cold sweats, night sweats); urinary (frequency, stress or urge incontinence); joint pains, and severe headaches/migraine [24], and whether they experienced those symptoms while working.

### 2.1. Outcome: Problems Coping with Menopause Symptoms at Work

Women who reported experiencing at least one of the menopausal symptoms while working were asked the following question “Do/did the menopausal symptoms cause you any problems coping with work?”. Possible answers were “no problems”, “minor problems”, “moderate problems”, and “severe problems”. Responses were combined in a binary variable so that those experiencing moderate or severe problems were compared with those experiencing minor or no problems.

### 2.2. Covariates

The baseline questionnaire included questions about: financial status (just about managing or worse vs. doing alright or better), having anyone financially dependent (yes vs. no), qualification level (school only/vocational training certificate/University degree or higher), marital status (single/widowed/divorced vs. married/partnership), smoking (ex/never vs. current), alcohol consumption (low/moderate/heavy drinker), weekly leisure time physical activity (none/up to 5 h/between 5 and 10 h/more than 10 h), self-rated health (fair/poor vs. at least good), depression (assessed using the Centre for Epidemiological Studies instrument scale [CES-D] and defined as CES-D ≥ 16) [25], and weight, which was combined with height to compute body mass index (BMI). Age was derived from date of birth and date of questionnaire completion. A series of job-related variables were also collected such as exposure to a list of occupational activities, shift, or night work (often vs. sometimes/never), and psychosocial work factors. Job title and industry were used to generate the nine major groups established in the Standard Occupational Classification (SOC) 2010 [26]: “Managers, directors and senior officials”, “Professional occupations”, “Associate professional and technical occupations”, “Administrative and secretarial occupations”, “Skilled trades occupations”, “Caring, leisure and other service occupations”, “Sales and customer service occupations”, “Process, plant and machine operatives”, “Elementary occupations”.

### 2.3. Statistical Analysis

To reduce recall bias, analysis was restricted to women who either started the menopause within the previous 10 years or who reported that they were currently experiencing symptoms (irrespective of when the menopause started). Socio-demographic and lifestyle characteristics as well as characteristics of the job were described according to whether participants experienced moderate/severe problems coping at work or not. Frequency and percentage distributions, means and standard deviations were used depending on the nature of the variable. Due to the relatively high prevalence of the outcome, we used Poisson model with the option for robust standard errors to explore the associations between risk factors and problems coping with work. Estimates of effect were therefore expressed as relative risk (RR) with the associated confidence interval (95% CI). After running unadjusted models, we retained all significant risk factors and fitted a final mutually adjusted model. All analyses were performed with Stata statistical software v17.0.

## 3. Results

At baseline, 4436 women were recruited to the HEAF study, amongst whom 3055 (69%) returned a usable questionnaire at follow-up 5 (2019). The 32 women who reported that they had not yet started the menopause were excluded. In total, 608 women reported starting the menopause within the preceding 10 years, amongst whom 471 were working at the time. Analyses were restricted to the 409 of these 471 women who were in work at the time of the baseline questionnaire and who did not change job over the course of the study, allowing us to be confident that the job they had when they experienced the menopause was the one reported at baseline. Approximately 27% of the sample reported at least moderate problems coping with the menopause at work. Figure 1 summarises the prevalence of different types of menopausal symptoms as reported by the 409 women, and the most common symptoms were: vasomotor (91.7%); trouble sleeping (68.2%); psychological (63.6%) and urinary (49.1%).

Table 1 describes the baseline demographic and lifestyle characteristics of the sample, stratified by whether women reported minimal problems coping with menopausal symptoms whilst working or moderate/severe problems. Age, educational attainment, marital status, BMI, and lifestyle factors were not different between women who reported problems coping as compared with women who did not. However, women who were struggling financially, had financial dependants, rated their own health poorly or were depressed according to their CES-D scores were more likely to report at least moderate problems coping.

The relationship between type of job, physical work demands, psychosocial work demands, and having difficulty with menopausal symptoms at work is summarised in Table 2. There were no differences by type of job or physical work characteristics or night/shift work. However, women who reported that their job was insecure, that were worried about their work, felt unappreciated or were dissatisfied with their job were more likely to report at least moderate problems coping with their menopausal symptoms at work.

Comparison of the prevalence of each type of menopausal symptoms by job type showed no clear patterns within or between groups. Likewise, when a particular symptom was reported, the proportion of women who described that they had problems coping at work because of their menopausal symptoms was generally similar. In contrast, comparison of the number of symptoms amongst those with no/minor problems (median = 4, IQR 2–5), as compared to those with moderate or severe problems (median = 6, IQR 4–7) showed higher numbers of symptoms generally amongst those who were most troubled (Figure 2).

Estimates adjusted for; difficulties with managing financially; having financial dependants; self-rated health; depression; job security; feeling angry about the job; job satisfaction; and feeling appreciated at work, as well as mutually adjusted, showed that the menopausal symptoms associated with the highest risk of reporting difficulty coping at work were severe headaches, joint aches and pains, and psychological symptoms (Figure 3).

## 4. Discussion

Amongst this contemporary cohort of over 400 British women working at the time of their menopause, almost one-third reported difficulty coping with their symptoms at work. Struggling with symptoms at work was not associated with lifestyle or demographic factors but instead with poorer financial circumstances, poorer self-rated health, depression, and with reporting higher numbers of symptoms. Our hypothesis that type of occupation might be associated with the likelihood of specific menopausal symptoms causing more difficulty in the workplace was not substantiated. We found similar prevalence rates for each symptom and similar proportions reporting difficulty coping across the range of occupational groups. Moreover, the impact of different menopausal symptoms did not appear to be affected by physical work demands. In contrast, psychosocial factors at work (job insecurity, worrying about work, job dissatisfaction, feeling poorly appreciated) were associated with an increased risk of menopausal symptoms impacting coping with work. After adjustment for all other factors, the three symptoms associated with the highest risk of reporting difficulty coping at work were: psychological symptoms (one or more of irritability, tearfulness, anxiety and depression), severe headaches and aches and pains in the joints.

Our finding that around one-third of working women report significant difficulty coping with their menopausal symptoms at work is consistent with prevalence rates reported in other studies. Albeit with some variation on the precise wording of the question, rates of 25% [27], 30% [14], 39.6% [21] and 40% [28] have been found elsewhere. As in the study by Woods and Mitchell, we found that poor self-perceived health was associated with more difficulty coping with these symptoms at work [29]. Given the increasing prevalence of women aged 45–55 years in the labour market, and the recognised association of symptomatic menopause with impaired productivity [6,9], disability for work [8,11,30], job exit and reduction in working hours [13], there is increasing evidence of the importance to employers of being aware of, and applying policy measures to, menopause amongst their workforce. In the current study, although elevated risk levels were found for short and longer-term (>20 days in past 12 months) sickness absence in those reporting most difficulties coping with their symptomatic menopause, neither attained statistical significance.

Our finding of an association between difficulty coping with menopausal symptoms at work and depression is not new. There is evidence from a number of longitudinal studies that some women are vulnerable to depressive disorder or symptoms during the menopause transition but that this risk attenuates 2–4 years after the final menstrual period [31]. Importantly, depression has been shown to be negatively correlated with performance at work, particularly among managers [14]. In the current study, the prevalence of psychological symptoms was around 60% in women doing each type of occupation and the proportions of workers with depression who reported difficulty coping at work was similarly elevated amongst women in professional occupations (47.3%) as women in elementary occupations (46.7%). The importance of number of symptoms to their impact at work has also been previously reported amongst Chinese working women, in whom increasing number of menopausal symptoms were also found associated with negative effects on self-rated health status [32].

An association between work psychosocial stressors and increased difficulties coping with menopausal symptoms at work has been previously reported [33]. Studies among teachers [17] and healthcare workers [18] found that they reported that work stress was the most important exacerbating factor for their menopausal symptoms. In another study, work stress was found to have negatively influenced menopausal symptoms [34]. Gujski et al. found that lacking social contacts at work, perceiving a lack of reward and perceived psychological demands of the work exacerbated stress amongst working menopausal women [35]. Qualitative researchers found that some women reported that they felt that they needed to hide their menopausal symptoms at work, so that being at work became a source of tension and anxiety [36]. Another qualitive study found that women who were asked about distress caused by vasomotor symptoms reported that being with male colleagues, or co-workers with whom they felt uncomfortable disclosing their menopause, exacerbated their problems coping at work [37]. In the reverse direction, Bariola et al. found that supervisor support and having control over work were associated with fewer symptoms [38]. We were unable to find other studies that had considered job security previously but insecurity has been found associated with both physical and mental health effects, including stress [39]. Im and colleagues previously reported a negative effect of menopausal symptoms on job satisfaction amongst immigrant Korean workers but their study, like the current one, was cross-sectional and it is unclear which is the direction of effect [40].

The current study suggested socio-economic differences in the menopause experience such that women reporting financial difficulties or dependants were more likely to report difficulty coping at work with symptoms. These results are similar to those reported amongst Chinese women in whom lower educational attainment and non-white collar occupation were reported as risk factors for more bothersome menopausal symptoms [32]. Educational attainment, employment status, and socio-economic position are strongly inter-linked and are importantly associated with many markers of general health such that those with poorer education tend to occupy lower-paid, more manual types of jobs and have poorer health status. Where other studies have compared the menopause experience amongst so-called “white collar” workers as compared with “blue collar” workers (traditionally, blue collar jobs were regarded as those involving manual work), they have found a higher prevalence of menopausal symptoms amongst women working in blue collar jobs than among those in white collar jobs [14]. They also found that the menopausal symptoms were more life-disrupting for blue collar than white collar workers and that some symptoms in particular were more negatively experienced in blue collar than white collar workers. The current study, which classified women’s jobs into nine occupational groups according to the SOC-10 could not find any major differences in prevalence of each type of symptom by type of occupation, nor were there any apparent differential effects of specific symptoms amongst women in different occupations. It may be that some of the previous findings about menopause experience are more reflective of socio-economic factors than of the jobs themselves. Certainly, we could find no association with any workplace physical demands, or the number of physically demanding tasks performed daily.

The findings from this study must be considered within the context of Hormone Replacement Therapy (HRT). It is recognised that HRT use is unequal in society such that women from higher social classes and those with higher levels of educational attainment are more likely to access HRT. In the current study approximately 25% participants reported ever having accessed HRT and ever using HRT was associated with an increased risk of reporting moderate/severe difficulties coping at work because of the menopausal symptoms (OR 1.54). Of course, this reflects reverse causation: those women who are most troubled by menopausal symptoms are those most likely to seek help and to be prescribed hormone therapy. Our finding of more severe work impacts amongst women with poorer socio-economic circumstances may at least partly reflect lack of access to HRT, perhaps because of a lack of health literacy or health self-efficacy and indeed our finding of less impact of menopausal symptoms amongst women with better socio-economic circumstances might reflect better access to appropriate HRT.

Some limitations of the study must be acknowledged. Firstly, the cross-sectional data collection about the menopause symptoms limits the possibility of inferring any causal association between variables. There is the possibility of recall bias as the median number of years since the menopause was seven. However, when we refitted the model on the sub-sample of women who reported that they were currently experiencing symptoms, we were reassured to find comparable results, which suggested recall bias did not play an important role. This was to our knowledge one of the first studies to investigate the effect of type of work on the experience of menopause. However, to do this, job titles have been classified into one of nine occupational groups. The SOC2010 classifies job titles into categories without considering job tasks, therefore women within the same classification might be doing quite different occupational tasks. For this reason, we also investigated the burden of menopausal symptoms according to number and type of self-reported physically demanding tasks. On the whole, individuals tend to over-estimate their physical demands at work when compared with formal occupational hygienist assessment, but this should not have affected internal comparisons summarised here. Finally, we used a menopausal symptom list that covered the domains described in the Greene climacteric scale [24] (symptoms not asked about were: excitability; loss of interest in sex; breathing difficulties). Women were not however asked to rate the severity of their symptoms.

## 5. Conclusions

A symptomatic menopause causes difficulties coping at work for around a third of working women. Women with depression and poorer financial circumstances are at increased risk and psychosocial, but not physical, work factors are associated. As recently discussed elsewhere [41], our findings also evidence the inequality that impacts working women’s menopause, suggesting that future workforce policy needs to be focused on supporting women who are doing the poorest paid jobs and have the greatest risk of poor health because of their deprived circumstances.

## Figures and Tables

**Figure 1 ijerph-20-00295-f001:**
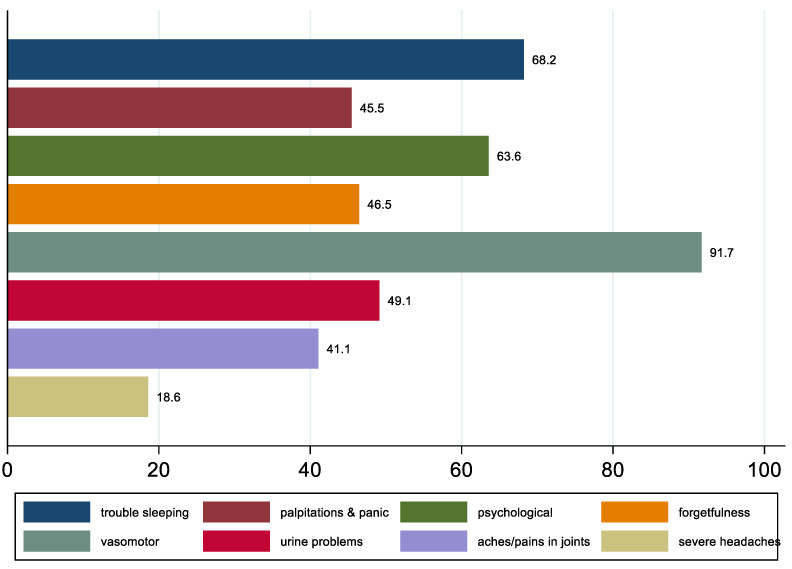
Prevalence of different types of menopausal symptoms among a population sample of 409 women who were working during their menopause.

**Figure 2 ijerph-20-00295-f002:**
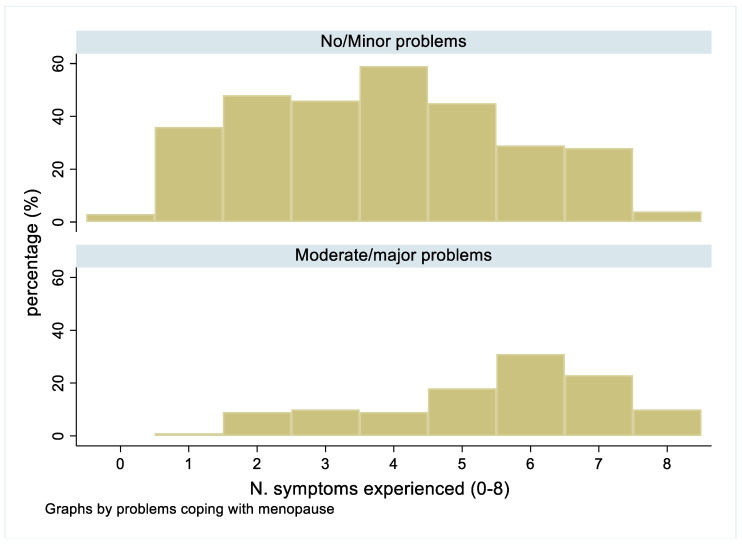
Comparison of the numbers of symptoms reported by women who reported no/minor problems coping with their symptoms at work as compared with those who reported moderate or major problems.

**Figure 3 ijerph-20-00295-f003:**
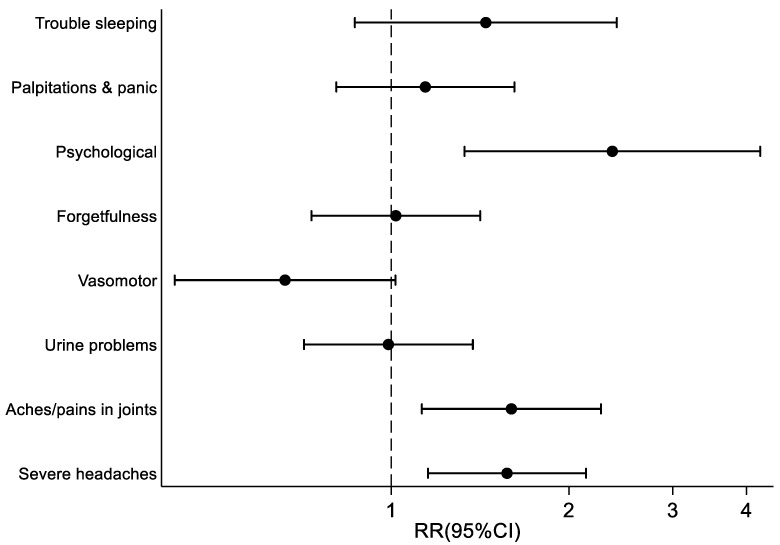
Relationship between individual menopausal symptoms and reporting difficulties coping with menopausal symptoms at work (Adjusted for all significant factors in Table 1 and Table 2 + mutual adjustment).

**Table 1 ijerph-20-00295-t001:** Relationship between baseline socio-demographic and lifestyle factors and reporting difficulties coping with menopausal symptoms at work amongst 409 women who were working at the time of their menopause.

	No/Minor Problems (*n* = 298)	Moderate/Major Problems (*n* = 111)	*p*-Value *	Crude Relative Risk RR (95% CI)
Age (years), mean (SD)	59.4 (2.4)	59.0 (2.2)	0.10	0.94 (0.87, 1.01)
Level of education				
No qualification/school only	93 (31.2)	32 (28.8)	0.27	0.83 (0.57, 1.20)
Vocational training certificate	77 (25.8)	22 (19.8)		0.72 (0.47, 1.11)
University degree/higher	128 (43.0)	57 (51.4)		Ref
Marital status				
Married/civil partnership	201 (67.5)	79 (71.2)	0.41	Ref
Single/divorced/widowed	93 (31.2)	32 (28.8)		0.91 (0.64, 1.29)
Missing	4 (1.3)	-		
Smoking				
Never/ex	275 (92.3)	102 (92.9)	0.40	Ref
Current	19 (6.4)	9 (8.1)		1.19 (0.67, 2.09)
Missing	4 (1.3)	-		
Alcohol consumption				
Low drinker	73 (24.5)	25 (22.5)	0.81	Ref
Moderate drinker	181 (60.7)	68 (61.3)		1.07 (0.72, 1.59)
Heavy drinker	17 (5.7)	9 (8.1)		1.36 (0.72, 2.54)
Missing	27 (9.1)	9 (8.1)		
Leisure-time physical activity				
None	50 (16.8)	22 (19.8)	0.79	Ref
Up to 5 h/week	177 (59.4)	67 (60.4)		0.90 (0.60, 1.35)
5–10 h/week	42 (14.1)	11 (9.9)		0.68 (0.36, 1.28)
More than 10 h/week	6 (2.0)	3 (2.7)		1.09 (0.41, 2.93)
Missing	23 (7.7)	8 (7.2)		
Body Mass Index (kg/m^2^)				
<18.5 (Underweight)	2 (0.7)	1 (0.9)	0.85	1.27 (0.25, 6.43)
18.5–24.9 (Normal)	135 (45.3)	48 (43.2)		Ref
25–29.9 (Overweight)	95 (31.9)	32 (28.8)		0.96 (0.65, 1.41)
≥30 (Obese)	59 (19.8)	26 (23.4)		1.17 (0.78, 1.74)
Missing	7 (2.4)	4 (3.6)		
Managing financially				
At least doing alright	213 (71.5)	61 (55.0)	0.002	Ref
Just about managing or worse	81 (27.2)	44 (39.6)		1.58 (1.14, 2.19)
Missing	4 (1.3)	6 (5.4)		
Anyone financially dependent				
No	265 (88.9)	86 (77.5)	0.005	Ref
Yes	29 (9.7)	19 (17.1)		1.62 (1.09, 2.40)
Missing	4 (1.3)	6 (5.4)		
Self-rated health				
At least good	260 (87.3)	87 (78.4)	0.08	Ref
Fair/poor	36 (12.1)	23 (20.7)		1.55 (1.08, 2.25)
Missing	2 (0.7)	1 (0.9)		
CESD score depression				
Not depressed	238 (79.9)	67 (61.5)	<0.001	Ref
Depressed	60 (20.1)	42 (38.5)		1.87 (1.37, 2.57)

* Pearson chi2 test for categorical variables; *t*-test for normally distributed variables.

**Table 2 ijerph-20-00295-t002:** Relationship between occupational characteristics and reporting difficulties coping with menopausal symptoms at work amongst 409 women who were working at the time of their menopause.

	No/Minor Problems (*n* = 298)	Moderate/Major Problems (*n* = 111)	*p*-Value *	Crude Relative Risk RR (95% CI)
**SOC2010 Major groups**				
Managers, directors and senior officials	23 (7.7)	7 (6.3)	0.35	0.71 (0.35, 1.43)
Professional occupations	82 (27.5)	40 (36.0)		Ref
Associate professional and technical occupations	30 (10.1)	14 (12.6)		0.97 (0.59, 1.60)
Administrative and secretarial occupations	75 (25.2)	20 (18.0)		0.64 (0.40, 1.02)
Skilled trades occupations	7 (2.4)	-		-
Caring, leisure and other service occupations	36 (12.1)	16 (14.4)		0.94 (0.58, 1.52)
Sales and customer service occupations	26 (8.7)	6 (5.4)		0.57 (0.27, 1.23)
Process, plant and machine operatives	4 (1.3)	1 (0.9)		0.61 (0.10, 3.59)
Elementary occupations	15 (5.0)	7 (6.3)		0.97 (0.50, 1.88)
**Occupational physical activities**				
Kneeling/squatting > 1 h/day	51 (17.1)	24 (21.6)	0.30	1.23 (0.84, 1.79)
Climbing ladder	17 (5.7)	6 (5.4)	0.91	0.96 (0.47, 1.95)
Digging	4 (1.3)	1 (0.9)	0.72	0.73 (0.13, 4.28)
Heavy lifting of 10+ kg	36 (12.1)	13 (11.7)	0.92	0.97 (0.59, 1.60)
Standing more than 3 h at a time	83 (27.9)	35 (31.5)	0.47	1.14 (0.81, 1.59)
Hard physical work that makes you feel sweaty	38 (12.8)	17 (15.3)	0.50	1.16 (0.76, 1.79)
**Number of physically demanding work activities**				
None	182 (61.1)	62 (55.9)	0.57	Ref
One	54 (18.1)	21 (18.9)		1.10 (0.72, 1.68)
Two or more	62 (20.8)	28 (25.2)		1.22 (0.84, 1.78)
**Working patterns**				
Often shift work	42 (14.1)	20 (18.0)	0.33	1.23 (0.82, 1.84)
Often night work	10 (3.4)	10 (9.0)	0.05	1.12 (0.99, 1.26)
**Psychosocial work factors**				
Feeling insecure	120 (40.3)	69 (62.2)	<0.001	1.91 (1.37, 2.66)
Often worry about job	43 (14.4)	39 (35.1)	<0.001	2.16 (1.59, 2.93)
Rarely/never feeling achievement	10 (3.4)	8 (7.2)	0.09	1.69 (0.98, 2.90)
Rarely/never feeling appreciated	21 (7.1)	15 (13.5)	0.03	1.20 (1.12, 1.30)
Often unfairly criticised	5 (1.7)	5 (4.5)	0.10	1.88 (0.99, 3.58)
Dissatisfied with job	11 (3.7)	13 (11.7)	0.002	2.13 (1.42, 3.19)

* Pearson chi2 test.

## Data Availability

The datasets used for this analysis are available on reasonable request from the MRC Versus Arthritis Centre for Musculoskeletal Health and Work by contacting Stefania D’Angelo: sd@mrc.soton.ac.uk.

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
