# Peer review of "Impact of Menopausal Symptoms on Work: Findings from Women in the Health and Employment after Fifty (HEAF) Study"

_ijerph, 2022, doi:10.3390/ijerph20010295_

Round 1

Reviewer 1 Report

Interesting study, brings new knowledge on factors related to menopausal symptoms on work. The study analyzed generally selected factors, therefore the conclusions were also general. I think that this work is a precursor to other works, from the beginning, focused on the study of factors related to menopausal symptoms on work.

Author Response

Thank you very much for your very positive feedback. We are pleased to hear you feel the study brings new knowledge about the effect that menopause has on working women. 

Reviewer 2 Report

The manuscript presents findings of research on the impact of menopausal symptoms on work studied by the standard methods. The paper presents very interesting results as well as an inquisitive and reliable interpretation of the research results.

Author Response

Thank you very much for your positive feedback about this manuscript.

Reviewer 3 Report

The manuscript “ Impact of menopausal symptoms on work: findings from women in the Health and Employment After Fifty (HEAF) 3 study”  was read with interest.

The problem of menopausal symptoms and work is important and the study seems well performed. However, after re-reading it is difficult to grasp and validate the message. Also, the statistical analysis is not clear.

It is hard to understand the outcome  Outcome: problems coping with menopause symptoms at work’ l 98.  Is this being troubled/disturbed by the symptoms or how women can handle the symptoms?

It surprises that hormone replacement therapy or another hormonal therapy is not mentioned.

The analysis is not clear

-        With a 0 to 3 scale, it is understandable to use binary data for making tables. However, for statistical analysis, I would rather expect to use the raw data indicating the severity of the symptoms

-        Intuitively one would expect that the severity and number or combination of symptoms would affect coping. Yet this typically multivariate analysis is not done. Which symptoms cluster? Is severity proportional in most women etc?

-        It can be argued whether the other variables as socioeconomic are co-variables of the model symptoms-coping.

Please consider these comments as an appreciation for the work done, and an incentive to improve the analysis and the writing. In its present form, the manuscript is not suitable for publication.

Author Response

The manuscript “ Impact of menopausal symptoms on work: findings from women in the Health and Employment After Fifty (HEAF) 3 study”  was read with interest.

The problem of menopausal symptoms and work is important and the study seems well performed. However, after re-reading it is difficult to grasp and validate the message. Also, the statistical analysis is not clear.

It is hard to understand the outcome  ‘Outcome: problems coping with menopause symptoms at work’ l 98.  Is this being troubled/disturbed by the symptoms or how women can handle the symptoms?

Thank you for this comment. We apologise if the previous version of the manuscript was not clear in respect to the definition of the outcome. Within the questionnaire, women who stated they experienced at least one of the symptoms of the menopause while in paid work, were asked the following question “Do/did the menopausal symptoms caused you any problems coping with work?”. Possible answers were “No problems”; “Minor problems”; “Moderate problems”; “Severe problems”. Please see revised version on page 3, lines 99-101.

It surprises that hormone replacement therapy or another hormonal therapy is not mentioned.

Many thanks for this very important comment. Within the questionnaire women were asked whether they had been taking HRT and for which length of time. Originally, we had performed analyses adjusted for this factor and we found that taking HRT was associated with difficulty coping with work. This is of course not surprising, and is explained by reverse causation: women who are most troubled by their menopausal symptoms are also those most likely to seek medical help and be prescribed treatment. We have added a section about this in the Discussion, page 10, lines 287-291

The analysis is not clear

We apologise for the lack of clarity. We have made some changes throughout the manuscript to improve it. In Table 1 and 2 we have added the word “crude” to specify that relative risks shown are unadjusted for confounders.  

We have also changed the title of Figure 3 to be consistent with titles of Table 1 and Table 2.

Furthermore, we have specified in the text (line 191-194, page 7) that estimates shown in Figure 3 are mutually adjusted as well as adjusted for the following covariates: managing financially, anyone financially dependent, self-rated health, depression, job security, feeling angry about the job, job satisfaction and feeling appreciated at work. 

-        With a 0 to 3 scale, it is understandable to use binary data for making tables. However, for statistical analysis, I would rather expect to use the raw data indicating the severity of the symptoms

Thank you for this comment. We agree with the reviewer that it would have been better to analyse the outcome as a 4-category variable. However, we were underpowered to conduct such analysis as only 15 women (3.7% of the sample) rated their ability to cope as “severe problems”. We therefore opted to combine categories of moderate or severe in order to reach a big enough sample (n=111, 27%).

-        Intuitively one would expect that the severity and number or combination of symptoms would affect coping. Yet this typically multivariate analysis is not done. Which symptoms cluster? Is severity proportional in most women etc?

Thank you for your comment. Unfortunately, our questionnaire did not assess the severity of symptoms, but only whether or not one or more of a pre-specified list of symptoms were experienced and then, whichever symptoms were reported, whether or not it was causing difficulties coping at work. We have added this as a limitation in line 309, page 10. We have computed the number of symptoms for each woman, but we felt that adding specific symptoms and number of symptoms in the same model would result in collinearity and biased the estimates. We have also explored whether symptoms would cluster together, however we did not get any meaningful cluster of symptoms with such analysis therefore we did not pursue this type of analysis any further.

-        It can be argued whether the other variables as socioeconomic are co-variables of the model symptoms-coping.

The reviewer is correct, it is indeed important to take into account socio-demographic and lifestyle factors in the association between menopausal symptoms and coping with work. We have adjusted for all socio-demographic, lifestyle and work factors that were significant in Table 1 and Table 2. We have clarified the list of confounders in page 7, lines 191-194.

Please consider these comments as an appreciation for the work done, and an incentive to improve the analysis and the writing. In its present form, the manuscript is not suitable for publication.

Thank you. We hope that the reviewer will now find the manuscript suitable for publication.

Round 2

Reviewer 3 Report

The manuscript “ Impact of menopausal symptoms on work: findings from women in the Health and Employment After Fifty (HEAF) study4” has improved a lot

Major problem

Notwithstanding the answers given, I still have problems with the conclusions. It cannot be excluded that higher and more educated social classes more frequently take JRT for their symptoms and thus cope better with work. It is suggested to add this in the conclusion or at least to add before “risk factors”, ‘Not taking HRT into account, risk….”

It is suggested also to add in the introduction that HRT is taken more by higher social classes and more educated women (CFR literature). And consistent with line 58 “although less so among 58 women who were managers [14]. (and probably took HRT). It could be explained that in this study this was a minority group and therefore HRT was not taken into account. (and thus will not show up in the statistics)

Fig 3 is surprising: I would expect vasomotor with a positive RR. Please check: it would make the data even more consistent  

Minor suggestion:

-        As a non-English native, I would replace in  the abstract  ‘coming at work’  with “coping with work” to emphasise it is not coping with the symptoms, but with work because of the symptoms.  

-        Explain statistics better L 135 unclear why Poisson model is better than non-Gaussian statistics  and   L127: univariate models ? “we retained all significant risk factors 

In conclusion; the authors are congratulated for a nice job, and the paper can be published provided these minor concerns are addressed.

Moreover, I would strongly suggest that the authors reanalyse their data taking into account

-        Symptoms: I would expect that urinary and the other symptoms cluster differently. If true a grouped index of the severity of “vasomotor (91.7%); trouble sleeping (68.2%); psychological (63.6%)” could become statistically much stronger especially if the severity of symptoms is used.  

-        HRT use:  more symptoms if no physical activity and poor … The higher social class might be less obese and thus taking more HRT but eliminated from this analysis

Author Response

Notwithstanding the answers given, I still have problems with the conclusions. It cannot be excluded that higher and more educated social classes more frequently take JRT for their symptoms and thus cope better with work. It is suggested to add this in the conclusion or at least to add before “risk factors”, ‘Not taking HRT into account, risk….”

It is suggested also to add in the introduction that HRT is taken more by higher social classes and more educated women (CFR literature). And consistent with line 58 “although less so among 58 women who were managers [14].”  (and probably took HRT). It could be explained that in this study this was a minority group and therefore HRT was not taken into account. (and thus will not show up in the statistics)

Please accept our apologies, we had not quite understood the important point that was being made. This paragraph has now been re-written to clarify this very point, as follows:

The findings from this study must be considered within the context of Hormone Replacement Therapy (HRT). It is recognised that HRT use is unequal in society such that women from higher social classes and those with higher levels of educational attainment are more likely to access HRT. In the current study approximately 25% participants reported ever having accessed HRT and ever using HRT was associated with an increased risk of reporting moderate/severe difficulties coping at work because of the menopausal symptoms (OR 1.54). Of course, this reflects reverse causation: those women who are most troubled by menopausal symptoms are those most likely to seek help and to be prescribed hormone therapy. Our finding of more severe work impacts amongst women with poorer socio-economic circumstances may be at least partly explained by lack of access to HRT, perhaps because of a lack of health literacy or health self-efficacy and indeed that perhaps our finding of less impact of menopausal symptoms amongst women with better socio-economic circumstances might reflect better access to appropriate HRT.

Fig 3 is surprising: I would expect vasomotor with a positive RR. Please check: it would make the data even more consistent 

 Thank you for the comment. We can assure the Reviewer that the data are correct. We suggest that this reflects the very high prevalence of vasomotor symptoms (91.7%) as shown in Figure 1, which means that the symptoms are not themselves very discriminatory in coping /not coping at work.

Minor suggestion:

-        As a non-English native, I would replace in  the abstract  ‘coming at work’  with “coping with work” to emphasise it is not coping with the symptoms, but with work because of the symptoms.  

      Thank you – we have made this change as suggested.

-        Explain statistics better L 135 unclear why Poisson model is better than non-Gaussian statistics  and   L127: univariate models ? “we retained all significant risk factors”  

In conclusion; the authors are congratulated for a nice job, and the paper can be published provided these minor concerns are addressed.

Thank you for your through review of our paper and interest in our work.

Moreover, I would strongly suggest that the authors reanalyse their data taking into account

-        Symptoms: I would expect that urinary and the other symptoms cluster differently. If true a grouped index of the severity of “vasomotor (91.7%); trouble sleeping (68.2%); psychological (63.6%)” could become statistically much stronger especially if the severity of symptoms is used.  

-        HRT use:  more symptoms if no physical activity and poor … The higher social class might be less obese and thus taking more HRT but eliminated from this analysis

Thank you for these suggestions. We will definitely consider drafting another paper with this different way of viewing our analyses.